# ACE: Exploring Activation Variance for Accurate and Calibration-Efficient LLM Pruning

## Abstract

With the rapid expansion of large language models (LLMs), the demand for memory and computational resources has grown significantly. Recent advances in LLM pruning aim to reduce the size and computational cost of these models. However, existing methods often suffer from either suboptimal pruning performance or low time efficiency during the pruning process. In this work, we propose an efficient and effective pruning method that simultaneously achieves high pruning performance and fast pruning speed with calibration efficiency. Our approach: (1) introduces an activation variance-guided pruning metric: a new metric that allows for better semantic information distinction preservation in the output activations after pruning; (2) enables model pruning with only a small sequence length of calibration dataset, while can maintain similar pruning performance as the original baselines that relies on larger sequence of calibration dataset (e.g. 2048 sequence lengths for Wanda and RIA). We conduct extensive experiments on prevalent LLMs, such as OPT, LLaMA, LLaMA-2, LLaMA-3, Qwen2.5, and MoE-based models such as Mixtral 8x7B. The experimental results show that we can achieve up to 18% decrease of perplexity and up to 63% less pruning time on WikiText-2, demonstrating the effectiveness of the proposed method.

## 1 Introduction

Recently, large language models (LLMs) have emerged as a prominent area of investigation, demonstrating exceptional capabilities through extensive parameterization across various tasks, such as language understanding (Devlin et al., 2018), text generation (Brown et al., 2020; Touvron et al., 2023a), question answering (Rajpurkar et al., 2016; Lewis et al., 2020), dialogue (Roller et al., 2021), and code generation (Chen et al., 2021), etc. While the increasing scale of LLMs has yield substantial accuracy inprovements, the advancement necessitates a compromise in memory consumption and inference latency (Devlin et al., 2019; Touvron et al., 2023a; Agarwal et al., 2023). For instance, deploying a LLaMA-65B model requires at least four A100-40GB GPUs, with the time-to-first-token (TTFT) exceeding 100 milliseconds (Yang et al., 2025), highlighting the significant limitations of practical deployment in resource-constrained environments. To mitigate the computational bottlenecks, various models compression techniques have been proposed, such as quantization (Bai et al., 2020; Frantar & Alistarh, 2022; Xiao et al., 2023; Lin et al., 2024), pruning (Wolff et al., 1992; LeCun et al., 1989; Mocanu et al., 2018; Sun et al., 2023; Frantar & Alistarh, 2023), weight decomposition (Hsu et al., 2022; Yang et al., 2024), etc. Among them, LLMs post-training pruning (Frantar & Alistarh, 2023; Sun et al., 2023) has garnered particular attention due to their ability in applying sparsity constraints to pre-trained LLMs without requiring computationally expensive retraining procedures, thus avoiding the prohibitive memory overhead.

Although existing LLMs post-training pruning methods (Sun et al., 2023; Frantar & Alistarh, 2023) have demonstrated potential in compressing model size with reduced memory overhead and negligible accuracy loss across diverse tasks, these approaches typically employ a well-designed weight importance evaluation metric with numerical magnitudes of weights and activations to identify important weight elements that should be preserved during pruning. In this work, *we identify a promising yet unexplored opportunities in designing the importance evaluation metrics via exploring the semantic information inherent in the input activation feature space*: **For equal-valued weights, those with lower input activation variance more effectively maintain token-level semantic distinctions:** previous studies (Ethayarajh, 2019; Gao et al., 2021b) have shown that reduced token-

level variation can result in semantic collapse and performance degradation across both classification and generation tasks. However, existing works fail to consider the variance of input activation features, a critical factor in preserving semantic distinctions. We observe that when comparing two same-valued weights, the weight associated with higher input activation produces reduced output activation differentiation across distinct tokens, thereby diminishing semantic distinctions. Consequently, such weights should be assigned lower importance scores compared to those exhibiting smaller variance, as they contribute less effectively to maintaining semantic diversity.

In this work, we propose ACE, which explores activation variance for accurate and calibration-efficient LLMs pruning. Inspired by the role of input activation feature variance, we design an activation variance-guided weight pruning metric (VarP), which incorporates a variance-based perturbation term and allows for better semantic information distinction preservation in the output activations after pruning. Moreover, we provide a theoretical analysis of the calibration efficiency of our approach and show that our method can achieve high accuracy even when applied with reduced sequence lengths for calibration data, demonstrating the potential of the proposed method in practical deployment scenarios with limited calibration data. Furthermore, VarP maintains (or even surpasses) the performance of full sequence length pruning baselines with fewer input sequence length and reduced pruning time, highlighting its effectiveness and calibration efficiency. We summarize our contributions as follows:

- We propose the activation variance-guided pruning metric, which includes the variance of input activation to avoid the diminish of distinction between different tokens during pruning.
- We theoretically analyze our proposed method can achieve calibration efficiency. Moreover, the experimental results demonstrate that our approach can achieve high accuracy on the pruned models with less input calibration sequence length and reduced pruning time.
- We conduct extensive experiments on various LLMs, such as OPT, LLaMA, LLaMA-2, LLaMA-3, Qwen2.5, and Mixtral-MoE models. Experimental results show that our method can outperform the baselines for both unstructured sparsity and N:M sparsity settings. For example, our VarP only takes 66% of the pruning time to perform 2:4 semi-structured pruning on Qwen2.5-32B compared to Wanda, while even obtaining about 0.4 reduction in perplexity compared with original Wanda.

## 2 RELATED WORK

**Network Pruning for Neural Networks.** Both unstructured pruning (Han et al., 2015; Frankle & Carbin, 2019) and structured pruning (Liu et al., 2017; Molchanov et al., 2019) are extensively explored for model compression and acceleration. The former identifies and removes individual weights based on criteria such as magnitude (Han et al., 2015) or gradient information (Lee et al., 2018). While achieving high sparsity, these methods often require specialized hardware to realize actual speedups. The latter focuses on removing entire structural components such as neurons, filters, or channels (Li et al., 2017; Liu et al., 2017). Among different structured sparsity patterns, the N:M sparsity (Mishra et al., 2021) has gained prominence, where N out of every M consecutive weights are retained. This pattern is adopted in NVIDIA's Ampere and later GPU architectures through specialized hardware support, enabling real-world efficient deployment and substantial acceleration during inference (Sun et al., 2023; Frantar et al., 2023; Zhang et al., 2024).

**Post-Training Pruning for Large Language Models.** Unlike training-aware sparsification (Gale et al., 2019), which iteratively prunes and fine-tunes the model during training, post-training pruning (PTP) operates directly on pretrained checkpoints, making it appealing for scenarios with limited training access or budget. However, designing effective pruning metrics remains a key challenge. Existing works such as Wanda (Sun et al., 2023) rely on the element-wise product of weight magnitudes and input activations to estimate importance. RIA (Zhang et al., 2024) incorporates relative importance between input and output channels to mitigate the problem of channel collapse. Pruner-zero (Dong et al., 2024) leverages evolutionary search to adaptively discover layer-wise metrics, while SparseGPT (Frantar & Alistarh, 2023) formulates pruning as a local reconstruction problem inspired by second-order approximations. However, current PTP works primarily focus on the enhancement of the element of the weights (Damadi, 2021; Dong et al., 2019; Sun et al., 2023; Frantar et al., 2023). Few studies explore the statistical information of input activations to further improve the pruning performance.

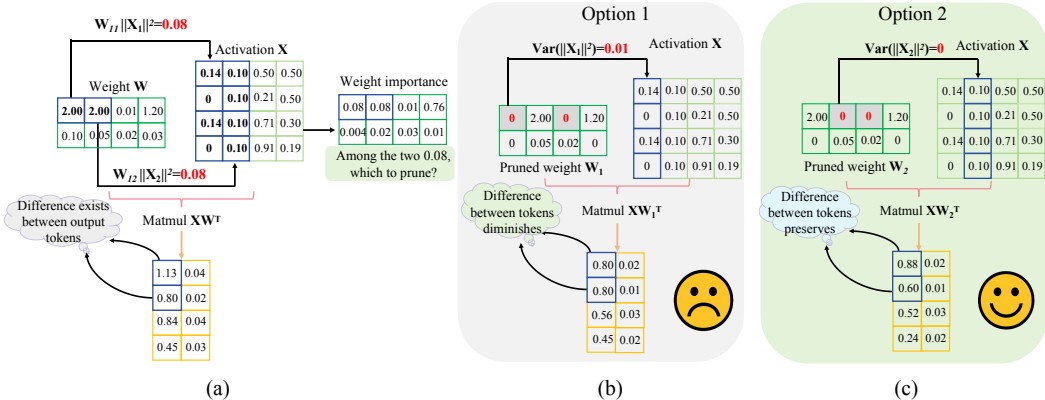

Figure 1: The motivating example of our proposed activation variance-guided pruning metric

# 3 METHODOLOGY

In this section, we will describe the motivation and our proposed ACE, an accurate and calibration efficient pruning approach of Large Language Models (LLMs). Firstly, we propose an activation variance-guided pruning metric which incorporates weights and the variance of input activations into the metric, aiming to maintain the relative distances between token representations in the embedding space during pruning, thus higher accuracy of the pruned model. Secondly, we theoretically analyze the calibration efficiency of our method.

## 3.1 ACTIVATION VARIANCE-GUIDED WEIGHT PRUNING METRIC (VARP)

**Motivation.** Most existing LLM pruning methods rely on importance metrics computed through various formulations involving weights and activations. However, the limitation arises when multiple elements share the same importance score. As illustrated in Figure 1(a), when two elements in the weight importance matrix in the first row have the same value (i.e., 0.08), it becomes difficult to determine which corresponding element to prune in the weight matrix $\mathbf{W}$ to achieve a target sparsity of 50%. Two different pruning options exist based on the choice of pruning weight elements with the same importance score. **Option 1** is to prune the first weight element in the first row with a corresponding larger variance (i.e., 0.01 is larger than 0), resulting in a reduced difference between the first two elements of first column in the output matmul compared to the original dense model, as illustrated in Figure 1(b). In contrast, **Option 2** prunes the second element in the first row of weight matrix, which better preserves the output disparities between the first two elements of the first column in the output, as shown in Figure 1(c), thereby maintaining closer alignment with the original distribution characteristics. For NLP tasks, preserving distinctions between output channels of tokens in the embedding space is crucial for maintaining semantic coherence and preventing the loss of token-level differences in model outputs (Ethayarajh, 2019; Li et al., 2020).

**VarP Design.** Motivated by the above example, we propose our activation variance-guided weight importance score metric which incorporates the variance of input activation as follows:

$$\mathbf{S}_{var_{ij}} = |\mathbf{W}_{ij}| \cdot (h(||\mathbf{X}_j||_2) + \text{Var}[||\mathbf{X}_j||_2^2]) \tag{1}$$

where $||\mathbf{X}_j||_2$ is the $l_2$ norm of $j$th features aggregated across $N$ different tokens, and $\text{Var}[||\mathbf{X}_j||_2^2]$ represents the variance of the squared values in the $j$-th column of the input activation. $h(\cdot)$ is used to represent the transformation of the $l_2$ norm and serve as a factor in the product of $|\mathbf{W}_{ij}|$ and $h(||\mathbf{X}_j||_2)$ to estimate the impact on the output when the weight element $\mathbf{W}_{ij}$ is removed. Simply, we can use $\mathbf{S}_{Wanda+var_{ij}}$ for deriving below. Building upon $h(||\mathbf{X}_j||_2)$ , we introduce an input variance-based perturbation term $\text{Var}[||\mathbf{X}_j||_2^2]$ to further determine which weight element should be pruned when the values of $|\mathbf{W}_{ij}| \cdot h(||\mathbf{X}_j||_2)$ are similar. We define $h(||\mathbf{X}_j||_2)$ as follows to evaluate the importance of the $j$th feature in input activation $\mathbf{X}$

$$h(||\mathbf{X}_j||_2) = (\mathbb{E}[||\mathbf{X}_j||_2^2])^2 + \mathbb{E}[||\mathbf{X}_j||_2^2] + 1 \tag{2}$$

where $\mathbb{E}[||\mathbf{X}_j||^2]$ represents the mean of the squared values in the $j$-th row of the input activation. Combining Equation 1 and Equation 2, we get our weight importance score metric as

$$\mathbf{S}_{var_{ij}} = |\mathbf{W}_{ij}| \cdot ((\mathbb{E}[||\mathbf{X}_j||_2^2])^2 + \text{Var}[||\mathbf{X}_j||_2^2] + \mathbb{E}[||\mathbf{X}_j||_2^2] + 1) \tag{3}$$

Based on the formula linking variance and expectation, we have

$$\text{Var}[X] = \mathbb{E}[X^2] - (\mathbb{E}[X])^2 \tag{4}$$

Combing Equation 3 and Equation 4, $\mathbf{S}_{var_{ij}}$ can be further derived as follows

$$\mathbf{S}_{var_{ij}} = |\mathbf{W}_{ij}| \cdot (\mathbb{E}[||\mathbf{X}_j||_2^4] + \mathbb{E}[||\mathbf{X}_j||_2^2] + 1) \tag{5}$$

Based on the fact that the input activations are normalized, implying that the corresponding activation values are less than 1. Then, based on the power series expansion

$$\mathbb{E}[||\mathbf{X}_j||_2^4] + \mathbb{E}[||\mathbf{X}_j||_2^2] + 1 = \mathbb{E}[||\mathbf{X}_j||_2^4 + ||\mathbf{X}_j||_2^2 + 1] \approx \mathbb{E}[\frac{1}{1 - ||\mathbf{X}_j||_2^2}] \tag{6}$$

We derive the importance score as

$$\mathbf{S}_{var_{ij}} = |\mathbf{W}_{ij}| \cdot \mathbb{E}[\frac{1}{1 - ||\mathbf{X}_j||_2^2}] \tag{7}$$

In the above equation, we use the absolute value of weight in the importance metric of our proposed method which we refer as *Wanda+VarP* in our experiments, since it shares the same weight component of the importance metric in Wanda (Sun et al., 2023). Similarly, we derive the importance metric of *RIA+VarP* as

$$\mathbf{S}_{RIA+var_{ij}} = \left( \frac{|\mathbf{W}_{ij}|}{\sum |\mathbf{W}_{*j}|} + \frac{|\mathbf{W}_{ij}|}{\sum |\mathbf{W}_{i*}|} \right) \cdot \mathbb{E}[\frac{1}{1 - ||\mathbf{X}_j||_2^2}] \tag{8}$$

where $\sum |\mathbf{W}_{*i}|$ and $\sum |\mathbf{W}_{j*}|$ denote the sum of the i-th row sum and the sum of j-th column in the weight matrix, respectively.

## 3.2 Calibration Data Efficiency Analysis

SparseGPT (Frantar & Alistarh, 2023) formulates LLMs post-training pruning as a layer-wise reconstruction problem, where for each layer, it aims to minimize the reconstruction error after pruning. Drawing inspiration from Optimal Brain Surgeon (OBS) (Hassibi & Stork, 1993), SparseGPT (Frantar & Alistarh, 2023) develops a pruning metric as follows

$$\mathbf{S}_{ij} = \frac{|\mathbf{W}_{ij}|^2}{\text{diag}\left((\mathbf{X}^T\mathbf{X} + \lambda\mathbf{I})^{-1}\right)_j}, \tag{9}$$

where $\mathbf{X}^T\mathbf{X} + \lambda\mathbf{I}$ represents the regularized Hessian matrix used in the layer-wise reconstruction problem and $\lambda$ is used to prevent algorithm failure due to singular matrices thus ensuring the Hessian is always invertible. Given the input activation as $\mathbf{X} = (\mathbf{X}_1, \mathbf{X}_2, \dots, \mathbf{X}_{d_{in}})$, we have

$$\frac{1}{\text{diag}\left((\mathbf{X}^T\mathbf{X} + \lambda\mathbf{I})^{-1}\right)_j} = \frac{\lambda + ||\mathbf{X}||^2}{\lambda + ||\mathbf{X}||^2 - \mathbf{X}_j^2} \cdot \lambda \tag{10}$$

Where $||\mathbf{X}||^2 = \sum_{i=1}^{d_{in}} \mathbf{X}_i^2$. Wanda (Sun et al., 2023) uses a coarse formulation to approximate $\text{diag}((\mathbf{X}^T\mathbf{X} + \lambda I)^{-1})$ as follows

$$\frac{1}{\text{diag}\left((\mathbf{X}^T\mathbf{X} + \lambda\mathbf{I})^{-1}\right)_j} \approx \frac{1}{(\text{diag}(\mathbf{X}^T\mathbf{X} + \lambda\mathbf{I}))_j^{-1}} = \mathbf{X}_j^2 + \lambda \tag{11}$$

Taking the difference of our derivation (i.e., Equation 10) and Wanda's approximation (i.e., Equation 11), we have

$$diff = |\frac{\lambda + ||\mathbf{X}||^2}{\lambda + ||\mathbf{X}||^2 - \mathbf{X}_j^2} \cdot \lambda - (\mathbf{X}_j^2 + \lambda)| \approx \frac{||\mathbf{X}||^2\mathbf{X}_j^2}{\lambda + ||\mathbf{X}||^2} \tag{12}$$

Suppose the input sequence length is denoted as $N$, we can further derive $diff$ as follows (Detailed derivation can be found at Appendix B.1.2)

$$diff = \frac{1}{N+1}\mathbb{E}[||\mathbf{X}_j||^2] \tag{13}$$

Equation 13 shows an inverse relationship between the sequence length $N$ and $diff$. Specifically, as $N$ decreases, $diff$ increases monotonically, which demonstrates that our proposed method yields reduced reconstruction error and improved accuracy, particularly in scenarios with smaller input sequence length. This theoretical finding suggests that our approach exhibits calibration data efficiency. The detailed derivation can be found in Appendix B.1.1 and B.1.2.

## 4 EXPERIMENTS

### 4.1 EXPERIMENTAL SETUP

**Models and Evaluations.** We use OPT 350M-13B (Zhang et al., 2022), LLaMA 7B-65B (Touvron et al., 2023a), LLaMA2 7B-13B (Touvron et al., 2023b), LLaMA 3 series (Grattafiori et al., 2024), Qwen2.5 1.5B-32B (Team, 2024), and MoE models such as Mixtral-8x7B and Mixtral-8x7B-Instruct (Jiang et al., 2024) to evaluate our proposed method. All model checkpoints used in our experiments are obtained from the HuggingFace Transformers library to ensure reproducibility. For fair comparison, we employ uniform pruning across all linear layers while preserving the embeddings and the head as dense (Sun et al., 2023; Zhang et al., 2024). We evaluate the proposed method in both generation task and zero-shot task. For the generation task, we measure the perplexity of the three model families on WikiText-2 (Merity et al., 2016). For zero-shot evaluation, we evaluate on seven benchmark tasks from EleutherAI LM Harness (Gao et al., 2021a) following existing work (Sun et al., 2023) on LLaMA models. All experiments are conducted on a server with 8 NVIDIA A100 GPUs, each with 40GB memory.

**Baselines.** Our baselines consist of two categories: one includes methods that only support 50% structured pruning such as SliceGPT (Ashkboos et al., 2024), SVD-LLM (Wang et al., 2024a), ASVD (Yuan et al., 2023), FLAP (An et al., 2024), SoBP (Wei et al., 2024) and CFSP (Wang et al., 2024b), and the other includes methods that not only support 50% structured pruning but also 2:4 and 4:8 semi-structured pruning methods, such as Wanda (Sun et al., 2023) and RIA (Zhang et al., 2024).

**Calibration Data.** For fair comparison with baselines, we take 128 samples from the C4 dataset (Raffel et al., 2020) for all models. Max context length size is used for both unstructured pruning and N:M semi-structured pruning for Wanda and RIA.

### 4.2 GENERATION TASK

We compare the VarP method with various LLM-based pruning baselines (e.g., SliceGPT (Ashkboos et al., 2024), SVD-LLM (Wang et al., 2024a), ASVD (Yuan et al., 2023), FLAP (An et al., 2024), SoBP (Wei et al., 2024) and CFSP (Wang et al., 2024b)) using the WikiText-2 dataset. We evaluated the performance in PPL of the LLaMA, LLaMA-2, LLaMA-3 and OPT model families in various sizes, as shown in Table 7 and Table 9. We also provide the time taken for the pruning process of the OPT and LLaMA models by these baselines, as shown in Table 12 and Table 10. The results of SliceGPT, SliceGPT-eq, SVD-LLM, ASVD, SoBP are from SoBP (Wei et al., 2024).

The experimental results show that compared to the baselines that only support structured pruning, our method outperforms them in both efficiency and performance at 50% sparsity. For example, on the LLaMA-7B model, our method achieves a PPL nearly 2.0 lower than SoBP, while running almost twice as fast as FLAP. Compared with the Wanda and RIA baselines, we can see that our method achieves results comparable to, or even better than, the original Wanda and RIA. For instance, on the LLaMA-7B model, our method achieves a PPL approximately 0.15 lower than the original Wanda. Moreover, our approach is much more time efficient than both the original Wanda and the RIA, taking only about 40% of the pruning time of the original Wanda on the OPT-30B model.

Table 1: PPL (↓) of LLaMA and LLaMA-2 models with VarP and baselines at 50% sparsity

| Method | LLaMA | | | | LLaMA-2 | |
|---|---|---|---|---|---|---|
| | 7B | 13B | 30B | 65B | 7B | 13B |
| SliceGPT | 15.94 | 9.79 | 8.22 | 6.92 | 12.80 | 10.60 |
| SliceGPT-eq | 46.08 | 11.89 | 9.89 | 8.10 | 16.02 | 13.38 |
| SVD-LLM | 13.85 | 10.22 | 7.96 | 6.69 | 16.14 | 10.79 |
| ASVD | 1.7e3 | 149.94 | 17.78 | 15.23 | 2.1e3 | 71.21 |
| FLAP | 20.80 | 13.60 | 9.59 | 7.05 | 21.94 | 13.70 |
| SoBP | 9.09 | 7.61 | 6.06 | 5.10 | 9.28 | 7.39 |
| CFSP | 10.18 | 8.32 | 7.06 | 6.25 | 9.31 | 8.00 |
| Wanda | 7.27 | 6.16 | 5.32 | 4.57 | 6.92 | 5.99 |
| RIA | **7.14** | **6.09** | **5.09** | **4.40** | 6.81 | 5.83 |
| **VarP (ours)** | 7.15 | 6.11 | 5.10 | 4.44 | **6.42** | **5.45** |

Table 2: Pruning time (s) (↓) of LLaMA and OPT models with VarP and baselines at 50% sparsity

| Method | LLaMA | | OPT | |
|---|---|---|---|---|
| | 7B | 13B | 6.7B | 13B |
| SliceGPT | 720 | 2400 | 840 | 2160 |
| SVD-LLM | 1440 | 7740 | 1980 | 9660 |
| ASVD | 6.2e4 | 3.9e6 | 7.5e4 | 3.7e5 |
| FLAP | 60 | 180 | 60 | 90 |
| SoBP | 1080 | 3900 | 2100 | 1.3e4 |
| CFSP | 40 | 58 | 43 | 62 |
| Wanda | 65 | 93 | 67 | 183 |
| RIA | 69 | 90 | 72 | 186 |
| **VarP (ours)** | **37** | **53** | **37** | **64** |

## 4.3 ZERO-SHOT TASKS

We report the zero-shot accuracy across seven tasks and the average accuracy of them on OPT-6.7B, LLaMA-7B, LLaMA-13B, and Qwen-2.5-3B from Table 3 to Table 6. We also have results on OPT-13B, LLaMA-2-7B, LLaMA-3.1-8B and LLaMA-2-13B, as shown from Table 15 to Table 18 in Appendix B.2.2. Across both unstructured and semi-structured sparsity settings, our method outperforms Wanda and RIA with sequences length of 16. VarP with the input sequence length of only 16 can generally surpass Wanda and RIA baselines pruning with 16 sequence lengths. For example, on the OPT-6.7B model with 2:4 semi-structured pruning, our method of RIA+VarP achieves an average accuracy of 46.46%, surpassing Wanda (Seq_Len = 16) by 1.1% and RIA (Seq_Len = 16) by 0.45%, respectively.

Table 3: Accuracy (↑) of OPT-6.7B on 7 zero-shot tasks

| Sparsity | Method | Seq_Len | BoolQ | RTE | HellaSwag | WinoGrande | ARC-e | ARC-c | OBQA | Avg. | Avg. Δ |
|---|---|---|---|---|---|---|---|---|---|---|---|
| 50% | Wanda | 16 | 62.32 | **53.43** | 46.03 | **61.48** | 62.66 | 27.22 | 24.60 | 48.24 | +1.19 |
| | Wanda+VarP (ours) | 16 | **66.64** | 53.42 | **47.65** | 60.85 | **63.17** | **28.85** | **25.40** | **49.43** | |
| | RIA | 16 | 63.97 | 52.71 | 46.69 | **61.33** | 62.88 | 27.90 | 24.40 | 48.55 | +0.87 |
| | RIA+VarP (Ours) | 16 | **66.29** | **53.09** | **47.69** | 61.04 | **63.74** | **28.28** | **25.80** | **49.42** | |
| 2:4 | Wanda | 16 | 62.17 | **52.35** | 40.91 | 59.74 | 56.06 | 24.74 | **21.60** | 45.36 | +0.53 |
| | Wanda+VarP (ours) | 16 | **62.35** | 51.26 | **42.23** | **60.93** | **57.70** | **26.19** | 20.60 | **45.89** | |
| | RIA | 16 | 62.19 | **53.42** | 41.29 | **61.01** | 56.88 | 25.50 | 21.80 | 46.01 | +0.45 |
| | RIA+VarP (Ours) | 16 | **63.57** | 51.64 | **42.68** | 59.99 | **58.13** | **26.38** | **22.80** | **46.46** | |
| 4:8 | Wanda | 16 | 62.22 | **53.41** | 43.45 | 60.13 | 58.95 | 26.71 | 23.00 | 46.83 | +0.11 |
| | Wanda+VarP (ours) | 16 | **63.94** | 52.71 | **45.45** | **61.48** | **60.19** | **26.87** | **24.80** | **47.92** | |
| | RIA | 16 | 63.15 | **53.79** | 43.98 | 60.77 | 59.05 | 26.88 | 23.60 | 47.31 | +1.30 |
| | RIA+VarP (Ours) | 16 | **64.80** | 52.35 | **45.55** | **61.72** | **61.20** | **27.48** | **25.20** | **48.61** | |

Table 4: Accuracy (↑) of LLaMA-7B on 7 zero-shot tasks

| Sparsity | Method | Seq_Len | BoolQ | RTE | HellaSwag | WinoGrande | ARC-e | ARC-c | OBQA | Avg. | Avg. Δ |
|---|---|---|---|---|---|---|---|---|---|---|---|
| 50% | Wanda | 16 | **70.73** | 54.51 | 51.51 | 64.87 | **69.48** | **36.09** | **29.00** | 53.74 | +0.69 |
| | Wanda+VarP (Ours) | 16 | 70.51 | **61.39** | **51.60** | **66.45** | 68.89 | 35.58 | 26.60 | **54.43** | |
| | RIA | 16 | **71.54** | 61.73 | 51.49 | **66.65** | **69.75** | **35.90** | **28.40** | 55.07 | +0.04 |
| | RIA+VarP (Ours) | 16 | 70.67 | **64.62** | **51.61** | 66.61 | 69.07 | 35.58 | 27.60 | **55.11** | |
| 2:4 | Wanda | 16 | **68.19** | 53.79 | 41.73 | 62.04 | 59.80 | 26.70 | **22.60** | 47.83 | +1.04 |
| | Wanda+VarP (Ours) | 16 | 68.02 | **54.88** | **43.89** | **63.61** | **61.32** | **28.83** | 21.60 | **48.87** | |
| | RIA | 16 | 67.98 | 55.23 | 42.03 | 62.03 | 60.48 | 26.96 | **24.20** | 48.41 | +0.96 |
| | RIA+VarP (Ours) | 16 | **68.87** | **56.68** | **43.88** | **63.06** | **61.74** | **28.59** | 22.80 | **49.37** | |
| 4:8 | Wanda | 16 | **70.00** | 55.23 | 46.81 | **64.09** | 63.38 | **31.91** | 24.80 | 50.88 | +0.79 |
| | Wanda+VarP (Ours) | 16 | 69.48 | **59.92** | **48.10** | 63.22 | **63.97** | 31.82 | **25.20** | **51.67** | |
| | RIA | 16 | 69.29 | 55.95 | 47.00 | **64.48** | 63.72 | 31.65 | **26.20** | 51.18 | +0.87 |
| | RIA+VarP (Ours) | 16 | **69.85** | **59.93** | **48.42** | 64.25 | **64.02** | **32.34** | 25.60 | **52.05** | |

## 4.4 PERFORMANCE ON LLMs WITH VARIOUS SCALES

Table 7 presents the performance of our method across different model scales, focusing on the OPT series models, ranging from 350M to 30B, evaluated on WikiText-2 datasets. Additional results for the Qwen2.5 series and Mixtral-MoE models on WikiText-2 are also provided in the Appendix B.2.1. We observe that our method is effective across models of varying scales. Compared with the baseline methods, our approach achieves performance comparable to, or even better than original Wanda, and

Table 5: Accuracy (↑) of LLaMA-13B comparison on 7 zero-shot tasks

| Sparsity | Method | Seq_Len | BoolQ | RTE | HellaSwag | WinoGrande | ARC-e | ARC-c | OBQA | Avg. | Avg. Δ |
|---|---|---|---|---|---|---|---|---|---|---|---|
| 50% | Wanda | 16 | **73.35** | 58.12 | **55.12** | 70.51 | **74.17** | 41.21 | **31.20** | **57.67** | -0.12 |
| | Wanda+VarP (Ours) | 16 | 73.33 | **59.94** | 54.78 | **70.80** | 73.10 | **41.56** | 29.40 | 57.55 | |
| | RIA | 16 | 73.34 | 57.11 | 54.51 | 70.62 | **73.94** | 40.87 | **30.60** | 57.28 | +0.03 |
| | RIA+VarP (Ours) | 16 | **73.98** | **58.48** | **54.97** | **70.64** | 72.65 | **41.47** | 29.00 | **57.31** | |
| 2:4 | Wanda | 16 | 70.12 | **53.79** | 46.44 | **66.45** | 65.82 | 32.25 | 25.80 | 51.52 | +0.47 |
| | Wanda+VarP (Ours) | 16 | **70.35** | 53.69 | **48.73** | 65.75 | 65.65 | **33.37** | **26.40** | **51.99** | |
| | RIA | 16 | 69.85 | **53.42** | 47.43 | **67.30** | 66.92 | 33.68 | 26.20 | 52.11 | +0.19 |
| | RIA+VarP (Ours) | 16 | **70.59** | 53.14 | **49.16** | 66.61 | 66.49 | **33.94** | 26.20 | **52.30** | |
| 4:8 | Wanda | 16 | 70.69 | 54.15 | 50.73 | **68.67** | 70.17 | 37.82 | **27.60** | 54.26 | +0.74 |
| | Wanda+VarP (Ours) | 16 | **72.94** | **54.51** | **52.45** | 68.35 | **70.67** | **38.48** | **27.60** | **55.00** | |
| | RIA | 16 | 71.16 | **53.43** | 51.24 | **70.48** | 69.95 | 37.37 | 28.00 | 54.52 | +0.45 |
| | RIA+VarP (Ours) | 16 | **72.68** | 53.09 | **52.57** | 67.62 | **71.07** | **38.99** | **28.80** | **54.97** | |

Table 6: Accuracy (↑) of Qwen2.5-3B on 7 zero-shot tasks

| Sparsity | Method | Seq_Len | BoolQ | RTE | HellaSwag | WinoGrande | ARC-e | ARC-c | OBQA | Avg. | Avg. Δ |
|---|---|---|---|---|---|---|---|---|---|---|---|
| 50% | Wanda | 16 | 66.11 | 72.34 | 45.40 | 64.25 | 71.51 | 37.29 | **26.80** | 54.81 | +0.88 |
| | Wanda+VarP (ours) | 16 | **68.70** | **73.29** | **45.52** | **66.23** | **71.72** | **37.63** | **26.80** | **55.69** | |
| | RIA | 16 | **67.89** | 76.17 | 45.81 | 65.19 | 71.75 | **38.31** | **27.80** | **56.13** | -0.09 |
| | RIA+VarP (Ours) | 16 | 67.40 | **76.43** | **45.92** | **65.90** | **72.15** | 37.73 | 26.80 | 56.04 | |
| 2:4 | Wanda | 16 | 63.21 | **63.18** | **35.77** | 58.01 | 56.39 | 24.74 | **20.00** | 45.90 | +0.02 |
| | Wanda+VarP (ours) | 16 | **64.83** | 56.31 | 35.75 | **58.73** | **59.39** | **26.45** | **20.00** | **45.92** | |
| | RIA | 16 | 63.85 | **60.28** | 36.52 | **58.63** | **60.52** | 27.21 | 18.20 | 46.45 | +0.29 |
| | RIA+VarP (Ours) | 16 | **64.10** | 60.13 | **38.30** | 57.83 | 59.85 | **27.38** | **19.60** | **46.74** | |
| 4:8 | Wanda | 16 | 62.54 | 61.01 | 41.45 | 60.77 | 66.07 | 31.14 | **23.60** | 49.51 | +0.93 |
| | Wanda+VarP (ours) | 16 | **63.37** | **65.71** | **41.87** | **61.88** | **66.33** | **31.74** | 22.20 | **50.44** | |
| | RIA | 16 | 62.32 | 58.48 | **42.19** | 61.32 | **67.34** | 31.56 | 22.20 | 49.34 | +1.27 |
| | RIA+VarP (Ours) | 16 | **62.91** | **66.43** | 41.80 | **62.27** | 66.09 | **32.00** | **22.80** | **50.61** | |

shows a notable PPL reduction compared with Wanda and RIA using 16 sequence lengths, with the improvement being particularly pronounced in semi-structured pruning for smaller models. For example, on the OPT-13B model with 2:4 structured pruning, the original Wanda achieves a PPL of 15.53, while Wanda with a sequence length of 16 achieves 15.34. In comparison, Wanda+VarP reaches approximately 13.95, representing a reduction of about 1.5. Similarly, RIA+VarP achieves a PPL that is approximately 1.6 lower than that of RIA with a sequence length of 16.

Table 7: PPL (↓) of OPT model series on WikiText-2 at different sparsity levels

| Sparsity | Method | Seq_Len | OPT | | | | | |
|---|---|---|---|---|---|---|---|---|
| | | | 350M | 1.3B | 2.7B | 6.7B | 13B | 30B |
| 50% | Wanda | 2048 | 36.24 | 18.40 | 14.22 | 11.98 | 11.92 | 10.03 |
| | Wanda | 16 | 42.91 | 24.62 | 17.62 | 14.33 | 12.34 | 11.01 |
| | Wanda+VarP (ours) | 16 | 37.78 | 20.77 | 15.02 | 12.32 | 11.64 | 10.29 |
| | RIA | 16 | 40.09 | 20.46 | 15.68 | 12.32 | 11.70 | 10.22 |
| | RIA+VarP (ours) | 16 | 37.40 | 19.90 | 14.70 | 12.08 | 11.38 | 10.13 |
| 2:4 | Wanda | 2048 | 114.57 | 28.15 | 21.27 | 15.91 | 15.53 | 13.47 |
| | Wanda | 16 | 136.12 | 35.63 | 25.69 | 18.58 | 15.34 | 19.13 |
| | Wanda+VarP (ours) | 16 | 103.80 | 29.59 | 24.45 | 16.11 | 13.95 | 14.02 |
| | RIA | 16 | 141.18 | 30.88 | 23.89 | 16.25 | 15.10 | 16.89 |
| | RIA+VarP (ours) | 16 | 120.04 | 28.06 | 23.48 | 15.84 | 13.54 | 13.43 |
| 4:8 | Wanda | 2048 | 58.95 | 22.20 | 16.78 | 13.55 | 13.38 | 10.87 |
| | Wanda | 16 | 66.70 | 27.82 | 20.55 | 17.00 | 13.28 | 12.58 |
| | Wanda+VarP (ours) | 16 | 58.13 | 23.34 | 17.61 | 13.77 | 12.33 | 11.13 |
| | RIA | 16 | 67.48 | 23.43 | 18.24 | 14.78 | 12.95 | 12.00 |
| | RIA+VarP (ours) | 16 | 63.14 | 22.36 | 17.57 | 13.62 | 12.06 | 10.95 |

## 4.5 PERFORMANCE ON MOE-BASED LLMS

We also have experiments to show the effectiveness of the proposed method on MoE-based models, such as the Mixtral-MoE 8×7B-v0.1 and Instruct models as shown in Table 8. Experimental results show that our method is effective for MoE models, particularly in the 2:4 and 4:8 semi-structured pruning settings. For instance, in Mixtral-MoE-8×7B-v0.1, our Wanda+VarP achieves a roughly 0.2 lower PPL compared to Wanda with a sequence length of 16, while RIA+VarP achieves a 0.3 lower PPL compared to RIA with a sequence length of 16.

## 4.6 CALIBRATION EFFICIENCY ANALYSIS ON DIFFERENT LLMS

Table 8: PPL (↓) of Mixtral-MoE model series on WikiText-2

| Sparsity | Method | Seq_Len | 8x7B-v0.1 | 8x7B-Instruct |
|---|---|---|---|---|
| 50% | Wanda | 2048 | 4.45 | 4.68 |
| | Wanda | 16 | 4.58 | 4.75 |
| | Wanda+VarP (ours) | 16 | 4.46 | 4.67 |
| | RIA | 16 | 4.52 | 4.63 |
| | RIA+VarP (ours) | 16 | 4.41 | 4.60 |
| 2:4 | Wanda | 2048 | 6.22 | 6.25 |
| | Wanda | 16 | 6.47 | 6.41 |
| | Wanda+VarP (ours) | 16 | 6.30 | 6.24 |
| | RIA | 16 | 6.30 | 6.43 |
| | RIA+VarP (ours) | 16 | 6.03 | 6.01 |
| 4:8 | Wanda | 2048 | 5.18 | 5.33 |
| | Wanda | 16 | 5.38 | 5.49 |
| | Wanda+VarP (ours) | 16 | 5.22 | 5.33 |
| | RIA | 16 | 5.25 | 5.40 |
| | RIA+VarP (ours) | 16 | 5.11 | 5.22 |

We conduct experiments to evaluate our proposed VarP approach, including the perplexity on WikiText-2 and runtime on LLaMA, LLaMA-2 and LLaMA-3 models in Table 9 and Table 10. We also provide more results of different models of pruning time such as Qwen2.5 and pruning time of Mixtral-MoE in Appendix B.2.1. Our key findings are summarized as follows:

Table 9: PPL (↓) of LLaMA, LLaMA-2, and LLaMA-3 families on WikiText-2

| Sparsity | Method | Seq_Len | LLaMA | | | | LLaMA-2 | | Meta-LLaMA-3 | LLaMA-3.1 | LLaMA-3.2 | |
|---|---|---|---|---|---|---|---|---|---|---|---|---|
| | | | 7B | 13B | 30B | 65B | 7B | 13B | 8B | 8B | 1B | 3B |
| 50% | Wanda | 2048 | 7.26 | 6.15 | 5.25 | 4.60 | 6.46 | 5.56 | 8.87 | 8.74 | 20.78 | 11.58 |
| | Wanda | 16 | 7.89 | 6.60 | 5.41 | 4.46 | 6.82 | 5.88 | 9.02 | 8.93 | 23.81 | 12.19 |
| | Wanda+VarP (ours) | 16 | 7.18 | 6.12 | 5.13 | 4.41 | 6.49 | 5.47 | 8.47 | 8.44 | 21.39 | 11.49 |
| | RIA | 16 | 7.27 | 6.11 | 5.14 | 4.42 | 6.44 | 5.53 | 8.35 | 8.34 | 19.63 | 11.32 |
| | RIA+VarP (ours) | 16 | 7.15 | 6.11 | 5.10 | 4.44 | 6.42 | 5.45 | 8.31 | 8.29 | 19.25 | 11.22 |
| 2:4 | Wanda | 2048 | 11.53 | 9.60 | 6.89 | 6.24 | 11.34 | 8.35 | 22.29 | 20.57 | 74.09 | 31.09 |
| | Wanda | 16 | 11.70 | 9.35 | 7.00 | 6.15 | 11.73 | 8.35 | 22.43 | 20.21 | 70.80 | 32.05 |
| | Wanda+VarP (ours) | 16 | 11.03 | 8.61 | 6.66 | 5.72 | 10.57 | 7.42 | 22.58 | 20.04 | 69.87 | 33.86 |
| | RIA | 16 | 11.53 | 8.83 | 6.81 | 6.05 | 11.17 | 7.90 | 22.62 | 20.33 | 89.69 | 34.74 |
| | RIA+VarP (ours) | 16 | 11.38 | 8.72 | 6.62 | 5.92 | 11.49 | 7.68 | 22.12 | 19.67 | 75.78 | 32.01 |
| 4:8 | Wanda | 2048 | 8.56 | 7.40 | 5.98 | 5.30 | 8.09 | 6.52 | 13.14 | 12.24 | 38.50 | 18.43 |
| | Wanda | 16 | 8.69 | 7.37 | 5.87 | 5.16 | 8.28 | 6.48 | 12.60 | 12.08 | 37.80 | 18.22 |
| | Wanda+VarP (ours) | 16 | 8.25 | 7.03 | 5.76 | 4.97 | 7.95 | 6.16 | 12.30 | 11.66 | 37.93 | 18.19 |
| | RIA | 16 | 8.46 | 7.14 | 5.78 | 5.08 | 8.05 | 6.34 | 12.20 | 11.75 | 41.06 | 18.93 |
| | RIA+VarP (ours) | 16 | 8.34 | 7.02 | 5.72 | 5.02 | 7.93 | 6.25 | 12.18 | 11.58 | 37.76 | 18.23 |

1) Compared to the full-sequence pruning variant of Wanda with 2048 sequence lengths, our VarP method offers substantial gains in time efficiency on LLaMA models. For example, in the 2:4 semi-structured pruning of the LLaMA-2-13B model, our method requires only about 25% of the pruning time compared to original Wanda. At the same time, Wanda+VarP achieves a PPL of 7.42, whereas both original Wanda and Wanda with a sequence length of 16 have a PPL of approximately 8.35, showing a reduction of around 1.0.

2) When compared to RIA with 16 sequence lengths, VarP exhibits slightly higher latency but yields significantly better evaluation performance. This accuracy gain is especially pronounced on the LLaMA-3 series of models. On the LLaMA-3.1-8B model with 2:4 semi-structured pruning, RIA with 16 sequence lengths produces a PPL of 20.33. In comparison, VarP can achieves a lower PPL of 19.67, representing a 0.7 reduction in perplexity and only have an additional latency about 16 seconds comparing to RIA with 16 sequence lengths.

3) While Wanda with 16 sequence lengths is the fastest among the methods, it comes at the cost of degraded performance. For example, on the LLaMA-7B model with 50% unstructured pruning, Wanda+VarP achieves a PPL of 7.18, outperforming Wanda (Seq_Len = 16) with PPL of 7.89.

Table 10: Pruning Time (s) (↓) of LLaMA, LLaMA-2, and LLaMA-3 model series on WikiText-2 at different sparsity levels

| Sparsity | Method | Seq_Len | LLaMA | | | | LLaMA-2 | | Meta-LLaMA-3 | LLaMA-3.1 | LLaMA-3.2 | |
| --- | --- | --- | --- | --- | --- | --- | --- | --- | --- | --- | --- | --- |
| | | | 7B | 13B | 30B | 65B | 7B | 13B | 8B | 8B | 1B | 3B |
| 50% | Wanda | 2048 | 64.6 | 39.4 | 49.3 | 67.6 | 98.8 | 183.5 | 76.7 | 74.8 | 28.9 | 52.4 |
| | Wanda | 16 | 23.7 | 29.8 | 40.0 | 60.7 | 22.7 | 27.9 | 30.4 | 29.8 | 15.2 | 25.3 |
| | Wanda+VarP (ours) | 16 | 29.9 | 37.4 | 53.1 | 78.4 | 29.2 | 36.7 | 44.4 | 43.8 | 22.7 | 35.6 |
| | RIA | 16 | 33.2 | 41.6 | 66.5 | 102.1 | 33.4 | 43.7 | 33.6 | 29.6 | 17.0 | 25.3 |
| | RIA+VarP (ours) | 16 | 47.2 | 52.2 | 82.6 | 120.7 | 45.0 | 57.6 | 45.7 | 44.6 | 24.1 | 37.5 |
| 2:4 | Wanda | 2048 | 80.6 | 121.6 | 260.9 | 463.7 | 179.4 | 267.7 | 118.9 | 117.7 | 39.3 | 74.0 |
| | Wanda | 16 | 41.1 | 55.7 | 98.3 | 152.7 | 42.7 | 55.3 | 73.9 | 74.2 | 26.2 | 49.8 |
| | Wanda+VarP (ours) | 16 | 48.3 | 65.9 | 110.6 | 174.4 | 48.2 | 65.2 | 86.4 | 86.0 | 33.0 | 61.4 |
| | RIA | 16 | 72.9 | 102.5 | 195.6 | 323.7 | 74.0 | 108.4 | 73.1 | 72.2 | 28.8 | 51.6 |
| | RIA+VarP (ours) | 16 | 84.9 | 118.3 | 212.7 | 342.5 | 88.6 | 116.1 | 88.2 | 88.8 | 34.9 | 64.1 |
| 4:8 | Wanda | 2048 | 70.2 | 104.2 | 226.8 | 410.3 | 160.2 | 234.6 | 95.9 | 96.7 | 33.8 | 62.5 |
| | Wanda | 16 | 31.0 | 40.2 | 65.9 | 100.9 | 31.3 | 40.1 | 50.6 | 52.2 | 21.0 | 37.4 |
| | Wanda+VarP (ours) | 16 | 38.1 | 57.2 | 80.8 | 121.5 | 38.4 | 50.5 | 62.6 | 61.4 | 27.4 | 50.2 |
| | RIA | 16 | 51.5 | 72.8 | 131.7 | 209.3 | 53.0 | 74.6 | 52.7 | 50.2 | 21.6 | 39.2 |
| | RIA+VarP (ours) | 16 | 67.9 | 92.8 | 148.2 | 228.6 | 67.1 | 86.0 | 64.5 | 66.3 | 29.5 | 51.2 |

## 4.7 IMPACT OF DIFFERENT SPARSITY

We report PPL of LLMs on WikiText-2 under different sparsity settings, as shown in Table 11. We can find that across different sparsity settings, our method, especially RIA+VarP, consistently matches or outperforms the baselines. At lower sparsity levels (20% and 40%), RIA sometimes performs better on certain OPT models but yields results comparable to ours on LLaMA models. As sparsity increases (50%–60%), our approach surpasses both RIA and Wanda in pruning effectiveness. Notably, on the

Table 11: PPL (↓) of LLMs with different sparsity on WikiText-2

| Sparsity | Method | Seq_Len | LLaMA | | LLaMA-2 | | OPT | | |
| --- | --- | --- | --- | --- | --- | --- | --- | --- | --- |
| | | | 7B | 13B | 7B | 13B | 1.3B | 6.7B | 13B |
| 20% | Wanda | 2048 | 5.81 | 5.13 | 5.22 | 4.68 | 14.69 | 10.62 | 10.06 |
| | Wanda | 16 | 5.76 | 5.13 | 5.17 | 5.09 | 15.32 | 10.43 | 9.94 |
| | Wanda+VarP (ours) | 16 | 5.72 | 5.11 | 5.14 | 4.62 | 15.09 | 10.94 | 10.16 |
| | RIA | 16 | 5.75 | 5.13 | **5.16** | 4.62 | **14.34** | **10.38** | **10.00** |
| | RIA+VarP (ours) | 16 | **5.74** | **5.12** | **5.16** | **4.59** | 14.96 | 10.93 | 10.16 |
| 40% | Wanda | 2048 | 6.39 | 5.51 | 5.66 | 5.01 | **15.87** | **10.96** | 10.64 |
| | Wanda | 16 | 6.48 | 5.64 | 5.75 | 5.09 | 18.57 | 12.13 | 10.66 |
| | Wanda+VarP (ours) | 16 | 6.27 | 5.48 | 5.63 | 4.95 | 17.52 | 11.29 | 10.62 |
| | RIA | 16 | 6.29 | 5.48 | **5.62** | 4.94 | **16.36** | **10.98** | **10.48** |
| | RIA+VarP (ours) | 16 | **6.23** | **5.47** | **5.62** | **4.92** | 16.96 | 11.24 | 10.58 |
| 60% | Wanda | 2048 | 10.69 | 8.75 | 10.04 | 7.93 | 26.53 | 15.21 | 15.94 |
| | Wanda | 16 | 13.35 | 9.75 | 11.34 | 8.60 | 45.77 | 20.32 | 17.52 |
| | Wanda+VarP (ours) | 16 | 11.02 | 8.37 | 11.11 | 7.21 | 30.56 | 16.24 | 14.42 |
| | RIA | 16 | 11.23 | 8.44 | 10.16 | 7.58 | 32.76 | 16.21 | 15.68 |
| | RIA+VarP (ours) | 16 | **10.82** | **8.16** | **10.02** | **7.28** | **28.37** | **15.04** | **14.15** |

OPT-13B model with 60% unstructured pruning, our method reduces PPL by about 1.9 compared to the baseline. Similarly, on the LLaMA-13B model under 60% unstructured pruning, we achieve a PPL of 8.16, outperforming Wanda's 8.75 (using a 2048-length input) by 0.6, while using much shorter sequences. In contrast, Wanda with shorter sequence lengths shows significant performance degradation, performing considerably worse than both our method and RIA.

## 5 CONCLUSION

This work introduces an activation variance-guided accurate and calibration-efficient post-training pruning technique tailored for large language models. We introduce VarP, an activation variance-guided weight pruning metric, which incorporates input activation variance into the pruning metric, achieving both pruning effectiveness and calibration efficiency. Through extensive experiments on prominent LLMs like OPT, LLaMA, LLaMA2, LLaMA3, Qwen2.5, and MoE-based Mixtral across varying model sizes, we show that VarP can achieve better performance than baselines with less pruning time and can combine with existing methods such as RIA for more efficient and accurate pruning. Moreover, experimental results show that our proposed LLM pruning method can be adapted to N:M sparsity and achieve better accuracy and calibration efficiency via taking benefit from the design of VarP.

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

## A    CLAIM OF LLM USAGE

In this work, large language models (LLMs) were used solely as a general-purpose writing assistant. Their role was limited to correcting grammar, fixing typographical errors, and polishing the language for clarity and readability.

## B    APPENDIX

### B.1    ACTIVATION VARIANCE-GUIDED WEIGHT PRUNING METRIC

#### B.1.1    DERIVATION OF VARP IMPORTANCE METRIC

Given the loss function $L(\mathbf{W})$, its Taylor expansion around the current weights $\mathbf{W}$ is:

$$L(\mathbf{W} + \Delta\mathbf{W}) \approx L(\mathbf{W}) + \nabla L(\mathbf{W})^T \Delta\mathbf{W} + \frac{1}{2}\Delta\mathbf{W}^T \mathbf{H}\Delta\mathbf{W} \tag{14}$$

where: - $\nabla L(\mathbf{W})$ is the gradient, - $\mathbf{H} = \nabla^2 L(\mathbf{W})$ is the Hessian (second derivative matrix). To minimize the loss variation after pruning, we focus on the second-order term $\frac{1}{2}\Delta\mathbf{W}^T \mathbf{H}\Delta\mathbf{W}$. However, calculating the full Hessian is computationally expensive for large models. SparseGPT approximates the Hessian using the input data matrix $\mathbf{X}$. Specifically, for MSE loss or linearized networks, $\mathbf{H} \approx \mathbf{X}^T\mathbf{X}$ and $\lambda\mathbf{I}$ is added as a regularization term to stabilize the inverse. Thus, the approximate local Hessian becomes:

$$\mathbf{H} \approx \mathbf{X}^T\mathbf{X} + \lambda\mathbf{I} \tag{15}$$

The pruning metric used in SparseGPT is:

$$\mathbf{S}_{ij} = \left[ \frac{|\mathbf{W}|^2}{\text{diag}\left((\mathbf{X}^T\mathbf{X} + \lambda\mathbf{I})^{-1}\right)} \right]_{ij} \tag{16}$$

which captures how important a weight is, normalized by the local curvature (second-order sensitivity) of the loss surface. If $\mathbf{X} \in \mathbb{R}^{1\times n}$ is an input: - $\mathbf{X}^T \in \mathbb{R}^{n\times 1}$, - $\mathbf{X}^T\mathbf{X} \in \mathbb{R}^{n\times n}$, a rank-1 matrix $\mathbf{u}\mathbf{v}^T$ where $\mathbf{u} = \mathbf{v} = \mathbf{X}^T$. Using the Sherman-Morrison formula, we have:

$$(\lambda\mathbf{I} + \mathbf{u}\mathbf{v}^T)^{-1} = \frac{1}{\lambda}\mathbf{I} - \frac{1}{\lambda^2}\frac{\mathbf{u}\mathbf{v}^T}{1 + \frac{\mathbf{v}^T\mathbf{u}}{\lambda}} \tag{17}$$

Specifically, the $i$-th diagonal element is:

$$\left[(\mathbf{X}^T\mathbf{X} + \lambda\mathbf{I})^{-1}\right]_{ii} = \frac{1}{\lambda} - \frac{1}{\lambda^2}\cdot\frac{\mathbf{X}_i^2}{1 + \frac{\|\mathbf{X}\|^2}{\lambda}} \tag{18}$$

where $\|\mathbf{X}\|^2 = \sum_{i=1}^n \mathbf{X}_i^2$. Then, we have :

$$\text{diag}\left((\mathbf{X}^T\mathbf{X} + \lambda\mathbf{I})^{-1}\right)_j = \frac{1}{\lambda} - \frac{\mathbf{X}_j^2}{\lambda^2 + \lambda\|\mathbf{X}\|^2} \tag{19}$$

Each diagonal element depends on the square of the corresponding feature in $X$. Then, we have:

$$\frac{1}{\text{diag}\left((\mathbf{X}^T\mathbf{X} + \lambda\mathbf{I})^{-1}\right)_j} = \frac{\lambda + \|\mathbf{X}\|^2}{\lambda + \|\mathbf{X}\|^2 - \mathbf{X}_j^2}\cdot\lambda \tag{20}$$

Note that the input $\mathbf{X}$ have been normalized, so $\|\mathbf{X}\|^2$ can be approximately viewed as a constant, we can denote it as $k$. The pruning metric can be simplified as:

$$\mathbf{S}_{ij} = \frac{|\mathbf{W}_{ij}|^2}{\text{diag}\left((\mathbf{X}^T\mathbf{X} + \lambda\mathbf{I})^{-1}\right)_j} = |\mathbf{W}_{ij}|^2 \cdot \frac{\lambda + k}{\lambda + k - \mathbf{X}_j^2}\cdot\lambda \tag{21}$$

Since $\lambda$ is a constant, we can omit it and we can get:

$$\mathbf{S}_{ij} = \frac{|\mathbf{W}_{ij}|^2}{\text{diag}\left((\mathbf{X}^T\mathbf{X} + \lambda\mathbf{I})^{-1}\right)_j} = |\mathbf{W}_{ij}|^2 \cdot \frac{1}{1 - \frac{\mathbf{X}_j^2}{\lambda+k}} \tag{22}$$

where $\lambda + k$ is a constant as well and we can omit it. We can use more input to calculate their mean value of calibration data and get our final pruning metric:

$$\mathbf{S}_{var_{ij}} = \mathbb{E}[\mathbf{S}_{ij}] = |\mathbf{W}_{ij}|^2 \cdot \mathbb{E}[\frac{1}{1 - ||\mathbf{X}_j||^2}] \tag{23}$$

### B.1.2 EFFICIENCY ANALYSIS

Now we analyze the efficiency of VarP. Wanda assumes the denominator of Equation 16 can be approximated:

$$\frac{1}{\text{diag}\left((\mathbf{X}^T\mathbf{X} + \lambda\mathbf{I})^{-1}\right)_j} \approx \frac{1}{(\text{diag}(\mathbf{X}^T\mathbf{X} + \lambda\mathbf{I}))_j^{-1}} = \mathbf{X}_j^2 + \lambda \tag{24}$$

Taking the difference of the right sides between Equation 20 and Equation 24, we have ($||\mathbf{X}||^2$ can be approximately replaced by $k$):

$$|\frac{\lambda(\lambda + k)}{\lambda + k - \mathbf{X}_j^2} - \mathbf{X}_j^2 - \lambda| = |\frac{\lambda}{1 - \frac{\mathbf{X}_j^2}{\lambda+k}} - \mathbf{X}_j^2 - \lambda| \tag{25}$$

Applying the power series expansion, we have:

$$|\frac{\lambda}{1 - \frac{\mathbf{X}_j^2}{\lambda+k}} - \mathbf{X}_j^2 - \lambda| = |\lambda(1 + \frac{\mathbf{X}_j^2}{\lambda+k}) - \mathbf{X}_j^2 - \lambda| = \frac{k\mathbf{X}_j^2}{\lambda + k} \tag{26}$$

If we have more sequences with length of $L$ as input, we can take the average of the $\mathbf{X}_j^2$, then we can the dfference between VarP and Wanda's as:

$$diff = \frac{k}{\lambda + k} \frac{\mathbf{X}_{j1}^2 + \mathbf{X}_{j2}^2 + \cdots + \mathbf{X}_{jL}^2}{L} = \frac{k}{\lambda + k}\mathbb{E}[||\mathbf{X}_j||^2] \tag{27}$$

Since the $\lambda$ is a constant, we can make it equals to $nk$, then:

$$diff = \frac{k}{\lambda + k}\mathbb{E}[||\mathbf{X}_j||^2] = \frac{1}{L+1}\mathbb{E}[||\mathbf{X}_j||^2] \tag{28}$$

We can see from the above Equation that if $L$ is small, the difference between these two methods will be large, which demonstrates the calibration efficiency of our proposed method.

### B.2 MORE EXPERIMENT RESULTS

### B.2.1 MORE RESULTS OF PPL AND PRUNING TIME ON MORE MODELS

Tables 12 provides the pruning time of OPT series models of our method comparing with baselines. Table 13 and Table 14 present the overall pruning time of the VarP method on the Qwen2.5 and Mistral-MoE models and PPL of Qwen2.5 models. It can be observed that, compared with the baseline, our method slightly increases the pruning time consumption. However, compared with the original baseline using a 2048 sequence length, the pruning time is significantly reduced. In experiments with Qwen2.5 and Mixtral-MoE models, we observe that our method performs similarly to the baselines under 50% sparsity. However, in semi-structured sparsity 2:4 and 4:8, our approach outperforms the baselines. For example, in the Qwen2.5-14B experiments, RIA+VarP achieves approximately a 0.3 reduction in PPL compared to RIA with 16 sequence lengths, while Wanda+VarP shows about a 0.5 reduction in PPL compared to Wanda with 16 sequence lengths. In terms of time, compared to Wanda with 2048 sequence lengths, the entire pruning process took only about 66% of Wanda's time while achieving an approximate 0.4 reduction in PPL.

Table 12: Pruning Time (s) (↓) of OPT model series on WikiText-2 at different sparsity levels

| Sparsity | Method | Seq_Len | OPT | | | | | |
|---|---|---|---|---|---|---|---|---|
| | | | 350M | 1.3B | 2.7B | 6.7B | 13B | 30B |
| 50% | Wanda | 2048 | 31.4 | 39.4 | 49.3 | 67.5 | 98.8 | 183.5 |
| | Wanda | 16 | 17.8 | 17.8 | 21.2 | 25.7 | 31.3 | 45.2 |
| | Wanda+VarP (ours) | 16 | 25.8 | 27.1 | 33.2 | 37.2 | 45.1 | 63.7 |
| | RIA | 16 | 16.1 | 19.8 | 21.8 | 24.4 | 31.6 | 44.8 |
| | RIA+VarP (ours) | 16 | 24.2 | 25.4 | 31.6 | 33.8 | 43.1 | 59.6 |
| 2:4 | Wanda | 2048 | 38.9 | 50.8 | 69.4 | 111.7 | 160.7 | 300.6 |
| | Wanda | 16 | 24.8 | 34.4 | 47.8 | 65.8 | 97.8 | 171.3 |
| | Wanda+VarP (ours) | 16 | 33.8 | 44.1 | 57.7 | 78.4 | 112.5 | 178.5 |
| | RIA | 16 | 23.1 | 32.4 | 46.8 | 63.8 | 95.7 | 161.5 |
| | RIA+VarP (ours) | 16 | 32.4 | 41.0 | 59.2 | 76.1 | 110.6 | 177.2 |
| 4:8 | Wanda | 2048 | 35.0 | 42.8 | 58.1 | 87.6 | 129.8 | 240.4 |
| | Wanda | 16 | 20.7 | 25.9 | 34.7 | 45.2 | 63.3 | 102.3 |
| | Wanda+VarP (ours) | 16 | 31.0 | 34.8 | 46.0 | 58.2 | 76.4 | 120.3 |
| | RIA | 16 | 20.2 | 24.5 | 34.8 | 45.5 | 64.5 | 101.9 |
| | RIA+VarP (ours) | 16 | 28.9 | 34.0 | 46.3 | 55.3 | 75.9 | 117.0 |

Table 13: PPL (↓) of Qwen-2.5 model series on WikiText-2

| Sparsity | Method | Seq_Len | 1.5B | 3B | 7B | 14B | 32B |
|---|---|---|---|---|---|---|---|
| 50% | Wanda | 2048 | 12.54 | 10.21 | 7.74 | 6.58 | 6.49 |
| | Wanda | 16 | 13.32 | 10.75 | 7.88 | 6.93 | 6.63 |
| | Wanda+VarP (ours) | 16 | 12.97 | 10.33 | 7.70 | 6.64 | 6.50 |
| | RIA | 16 | 12.46 | 9.98 | 7.59 | 6.51 | 6.50 |
| | RIA+VarP (ours) | 16 | 12.42 | 9.94 | 7.55 | 6.49 | 6.46 |
| 2:4 | Wanda | 2048 | 42.30 | 21.16 | 13.10 | 12.37 | 8.90 |
| | Wanda | 16 | 43.96 | 23.41 | 13.55 | 11.53 | 8.71 |
| | Wanda+VarP (ours) | 16 | 42.02 | 23.73 | 12.93 | 10.98 | 8.45 |
| | RIA | 16 | 45.46 | 22.11 | 12.53 | 11.28 | 8.59 |
| | RIA+VarP (ours) | 16 | 45.20 | 21.64 | 12.48 | 11.02 | 8.32 |
| 4:8 | Wanda | 2048 | 19.39 | 13.67 | 9.34 | 8.38 | 7.32 |
| | Wanda | 16 | 20.35 | 14.13 | 9.58 | 7.97 | 7.45 |
| | Wanda+VarP (ours) | 16 | 20.13 | 13.99 | 9.40 | 7.80 | 7.35 |
| | RIA | 16 | 20.37 | 13.54 | 9.18 | 7.77 | 7.33 |
| | RIA+VarP (ours) | 16 | 20.03 | 13.47 | 9.18 | 7.59 | 7.14 |

### B.2.2 MORE RESULTS ON ZERO-SHOT TASKS

Tables 15 to Table 18 present more zero-shot evaluation accuracy of LLaMA models and OPT models. It can be observed that our method generally outperforms the baseline methods in evaluation accuracy. For example, in the 4:8 semi-structured pruning task of the OPT-13B model, our method achieves nearly a 2% higher average evaluation accuracy compared to the baseline.

### B.2.3 ROBUSTNESS ANALYSIS

To evaluate the robustness and stability of our method, we conduct experiments across multiple calibration data sampling configurations We select three different random seeds, therefore three different calibration data subsets, and repeat the pruning process for each of them on LLaMA-7B model using WikiText-2 dataset. It's shown in Table 19 that our method exhibits better results across different random seeds.

Table 14: Pruning time (s) (↓) of Qwen-2.5 and Mixtral-MoE model series on WikiText-2 at different sparsity levels

| Sparsity | Method | Seq_Len | Qwen-2.5 | | | | | Mixtral-MoE | |
|---|---|---|---|---|---|---|---|---|---|
| | | | 1.5B | 3B | 7B | 14B | 32B | 8x7B-v0.1 | 8x7B-Instruct |
| 50% | Wanda | 2048 | 28.9 | 40.6 | 74.7 | 125.9 | 218.2 | 174.3 | 180.8 |
| | Wanda | 16 | 16.7 | 20.8 | 17.5 | 46.0 | 64.8 | 99.3 | 99.8 |
| | Wanda+VarP (ours) | 16 | 22.7 | 28.8 | 24.0 | 66.6 | 95.8 | 159.3 | 160.1 |
| | RIA | 16 | 19.8 | 20.6 | 17.6 | 46.8 | 65.6 | 102.2 | 103.4 |
| | RIA+VarP (ours) | 16 | 23.4 | 27.9 | 25.4 | 67.2 | 96.3 | 160.5 | 161.8 |
| 2:4 | Wanda | 2048 | 36.3 | 54.2 | 113.6 | 197.8 | 353.5 | 410.0 | 410.0 |
| | Wanda | 16 | 25.8 | 33.2 | 36.9 | 122.0 | 203.8 | 332.9 | 330.0 |
| | Wanda+VarP (ours) | 16 | 32.1 | 41.9 | 42.6 | 147.2 | 238.6 | 387.2 | 389.5 |
| | RIA | 16 | 25.0 | 33.9 | 37.8 | 121.9 | 205.3 | 335.3 | 331.5 |
| | RIA+VarP (ours) | 16 | 31.5 | 43.6 | 45.8 | 146.3 | 236.3 | 390.1 | 391.4 |
| 4:8 | Wanda | 2048 | 32.4 | 46.2 | 93.8 | 159.7 | 285.2 | 292.5 | 295.7 |
| | Wanda | 16 | 20.6 | 26.5 | 26.3 | 83.5 | 131.4 | 214.6 | 212.5 |
| | Wanda+VarP (ours) | 16 | 28.0 | 35.0 | 33.5 | 106.4 | 146.5 | 271.0 | 273.4 |
| | RIA | 16 | 21.0 | 26.7 | 28.6 | 82.4 | 132.5 | 217.4 | 214.6 |
| | RIA+VarP (ours) | 16 | 28.0 | 34.7 | 35.4 | 105.5 | 164.5 | 275.3 | 277.2 |

Table 15: Accuracy (↑) of OPT-13B on 7 zero-shot tasks

| Sparsity | Method | Seq_Len | BoolQ | RTE | HellaSwag | WinoGrande | ARC-e | ARC-c | OBQA | Avg. | Avg. Δ |
|---|---|---|---|---|---|---|---|---|---|---|---|
| 50% | Wanda | 16 | **65.75** | 53.43 | 48.39 | 62.66 | 64.10 | 29.78 | **26.20** | 50.04 | +1.32 |
| | Wanda+VarP (Ours) | 16 | 65.11 | **56.68** | **50.23** | **63.70** | **65.90** | **31.74** | 26.20 | **51.36** | |
| | RIA | 16 | **65.86** | 54.11 | 49.01 | 62.84 | 64.31 | 30.90 | **26.60** | 50.52 | 0.00 |
| | RIA+VarP (Ours) | 16 | 63.00 | **55.24** | **49.94** | **63.14** | **65.49** | **31.91** | 25.00 | 50.52 | |
| 2:4 | Wanda | 16 | **64.31** | 52.70 | 43.80 | 60.93 | 59.55 | 26.11 | 22.60 | 47.14 | +0.97 |
| | Wanda+VarP (Ours) | 16 | 58.93 | **53.07** | **47.31** | **63.07** | **61.53** | **28.67** | **24.20** | 48.11 | |
| | RIA | 16 | **65.65** | 52.70 | 44.57 | 61.88 | 58.63 | 27.05 | 21.60 | 47.44 | +0.66 |
| | RIA+VarP (Ours) | 16 | 62.14 | **53.79** | **46.78** | **62.51** | **60.86** | **28.07** | **22.60** | **48.10** | |
| 4:8 | Wanda | 16 | **65.29** | 53.07 | 46.16 | 62.66 | 60.81 | 27.30 | 24.80 | 48.58 | +1.97 |
| | Wanda+VarP (Ours) | 16 | 62.30 | **59.57** | **48.74** | **63.30** | **63.68** | **31.05** | **25.20** | 50.55 | |
| | RIA | 16 | **65.44** | 52.34 | 46.54 | **63.38** | 60.86 | 27.64 | 24.80 | 48.71 | +1.11 |
| | RIA+VarP (Ours) | 16 | 61.45 | **57.77** | **48.47** | 62.83 | **62.96** | **30.29** | 25.00 | **49.82** | |

Table 16: Accuracy (↑) of LLaMA-3.1-8B on 7 zero-shot tasks

| Sparsity | Method | Seq_Len | BoolQ | RTE | HellaSwag | WinoGrande | ARC-e | ARC-c | OBQA | Avg. | Avg. Δ |
|---|---|---|---|---|---|---|---|---|---|---|---|
| 50% | Wanda | 16 | 75.90 | **61.01** | **51.30** | 67.88 | 70.87 | 39.57 | **27.20** | **56.24** | -0.03 |
| | Wanda+VarP (ours) | 16 | **76.55** | 56.31 | 51.23 | **69.22** | **73.12** | **40.28** | 26.80 | 56.21 | |
| | RIA | 16 | 77.40 | **56.31** | 51.53 | **69.21** | 72.81 | **41.46** | **28.20** | **56.70** | -0.31 |
| | RIA+VarP (ours) | 16 | **78.21** | 55.23 | **51.73** | 69.14 | 72.72 | 40.96 | 26.80 | 56.39 | |
| 2:4 | Wanda | 16 | 64.61 | **54.51** | 37.31 | 59.74 | 58.20 | 25.25 | **19.00** | 45.52 | +0.63 |
| | Wanda+VarP (ours) | 16 | **66.62** | 52.35 | **38.70** | **59.98** | **60.32** | **26.46** | 18.60 | **46.15** | |
| | RIA | 16 | **66.54** | 53.07 | 38.31 | 59.27 | **59.97** | 28.32 | 19.40 | 46.41 | +0.09 |
| | RIA+VarP (ours) | 16 | 66.34 | **53.09** | **38.39** | **59.89** | 59.81 | **28.62** | 19.40 | **46.50** | |
| 4:8 | Wanda | 16 | 66.72 | 53.43 | 44.10 | **66.22** | 65.36 | 30.80 | 23.40 | 50.00 | +0.62 |
| | Wanda+VarP (ours) | 16 | **67.34** | **53.80** | **44.83** | 65.68 | **65.41** | **33.12** | **24.20** | **50.62** | |
| | RIA | 16 | 67.72 | **53.42** | **44.48** | 65.11 | 64.84 | 32.24 | **25.00** | 50.40 | +0.08 |
| | RIA+VarP (ours) | 16 | **68.32** | 53.07 | 44.39 | 64.96 | **65.38** | **33.20** | 24.00 | **50.48** | |

Table 17: Accuracy (↑) of LLaMA-2-7B on 7 zero-shot tasks

| Sparsity | Method | Seq_Len | BoolQ | RTE | HellaSwag | WinoGrande | ARC-e | ARC-c | OBQA | Avg. | Avg. Δ |
|---|---|---|---|---|---|---|---|---|---|---|---|
| 50% | Wanda | 16 | **74.44** | 55.59 | **51.29** | 66.12 | **71.63** | 37.25 | **29.40** | **55.10** | -0.18 |
| | Wanda+VarP (Ours) | 16 | 73.70 | **57.76** | 51.01 | **66.15** | 70.38 | **37.30** | 28.20 | 54.92 | |
| | RIA | 16 | 73.70 | 55.59 | 51.37 | **66.45** | 69.69 | 35.66 | 28.60 | 54.43 | +0.44 |
| | RIA+VarP (Ours) | 16 | **73.72** | **55.62** | **51.53** | 66.08 | **70.48** | **37.47** | **29.20** | **54.87** | |
| 2:4 | Wanda | 16 | **67.83** | 53.43 | 39.85 | 59.59 | 59.30 | 27.04 | **21.80** | 46.98 | +0.73 |
| | Wanda+VarP (Ours) | 16 | 65.85 | **53.45** | **42.18** | **60.93** | **63.55** | **28.41** | 19.60 | **47.71** | |
| | RIA | 16 | **66.90** | 53.39 | 40.27 | 59.59 | 60.77 | 27.98 | **21.80** | 47.24 | +0.65 |
| | RIA+VarP (Ours) | 16 | 65.34 | **54.20** | **42.45** | **61.29** | **62.54** | **28.62** | 20.80 | **47.89** | |
| 4:8 | Wanda | 16 | **71.56** | **54.15** | 45.29 | 63.69 | 65.48 | 32.51 | **24.80** | 51.07 | +0.32 |
| | Wanda+VarP (Ours) | 16 | 70.33 | 53.80 | **46.80** | **64.71** | **66.29** | **34.39** | 23.40 | **51.39** | |
| | RIA | 16 | **72.88** | 54.14 | 45.82 | 64.15 | 66.27 | 32.75 | **24.80** | 51.54 | +0.14 |
| | RIA+VarP (Ours) | 16 | 69.70 | **54.51** | **47.30** | **64.72** | **66.37** | **34.56** | 24.60 | **51.68** | |

Table 18: Accuracy (↑) of LLaMA-2-13B on 7 zero-shot tasks

| Sparsity | Method | Seq_Len | BoolQ | RTE | HellaSwag | WinoGrande | ARC-e | ARC-c | OBQA | Avg. | Avg. Δ |
|---|---|---|---|---|---|---|---|---|---|---|---|
| 50% | Wanda | 16 | 78.66 | **62.45** | 56.61 | **70.56** | 76.72 | **42.58** | 32.00 | **59.94** | -0.31 |
| | Wanda+VarP (Ours) | 16 | **78.84** | 62.10 | **56.97** | 69.54 | 75.80 | 41.80 | **32.40** | 59.63 | |
| | RIA | 16 | 79.90 | **62.09** | 56.31 | 70.00 | 76.18 | 40.27 | 31.00 | 59.39 | +0.72 |
| | RIA+VarP (Ours) | 16 | **80.57** | 60.94 | **57.49** | **70.16** | **76.94** | **42.49** | **32.20** | **60.11** | |
| 2:4 | Wanda | 16 | **77.42** | **59.55** | 46.01 | **66.92** | 68.57 | 33.10 | 23.60 | 53.59 | +0.32 |
| | Wanda+VarP (Ours) | 16 | 77.03 | 56.32 | **48.74** | 65.12 | **69.02** | **36.18** | **25.00** | **53.91** | |
| | RIA | 16 | **77.18** | **58.84** | 46.83 | **66.69** | 68.77 | 33.27 | 23.40 | 53.57 | +0.49 |
| | RIA+VarP (Ours) | 16 | 76.97 | 58.42 | **49.07** | 64.03 | **69.79** | **34.97** | **25.20** | **54.06** | |
| 4:8 | Wanda | 16 | **79.35** | 60.29 | 51.67 | 68.03 | **74.03** | **38.90** | 27.80 | 57.15 | +0.37 |
| | Wanda+VarP (Ours) | 16 | 78.78 | **61.73** | **53.45** | **68.48** | 72.89 | 38.57 | **28.80** | **57.52** | |
| | RIA | 16 | **79.75** | 60.29 | 51.85 | **68.51** | 73.10 | **38.31** | 27.00 | 56.97 | +0.43 |
| | RIA+VarP (Ours) | 16 | 77.49 | **62.77** | **53.63** | 67.94 | **73.18** | 37.60 | **29.20** | **57.40** | |

Table 19: Perplexity for pruned LLaMA-7B models with different random seeds on WikiText-2

| Model | Method | seed#1 | seed#2 | seed#3 |
|---|---|---|---|---|
| LLaMA-7B | Wanda-16 | 7.89 | 8.19 | 7.76 |
| | RIA with 16 sequence lengths | 7.27 | 7.28 | 7.26 |
| | Wanda+VarP (Ours) | **7.18** | **7.18** | **7.120** |

Table 20: Perplexity of pruned LLaMA-7B model with different sequence lengths on WikiText-2

| Model | Method | Sequence Lengths | | | | | |
|---|---|---|---|---|---|---|---|
| | | 16 | 32 | 128 | 512 | 1024 | 2048 |
| LLaMA-7B | Wanda | 7.89 | 7.52 | 7.28 | 7.27 | 7.27 | 7.26 |
| | RIA | 7.27 | 7.22 | 7.15 | 7.13 | 7.12 | 7.12 |
| | Wanda+VarP (Ours) | 7.18 | 7.19 | 7.20 | 7.25 | 7.26 | 7.25 |

### B.2.4 IMPACT OF SEQUENCE LENGTHS

We present the impact of sequence lengths on the LLaMA-7B model for WikiText-2 under 50% unstructured sparsity in Table 20. It's shown that our method consistently achieves better pruning performance, especially when the sequence length is relatively small, yielding much lower perplexity compared to both RIA and Wanda. This aligns with our theoretical analysis: with the decreasing of the input sequence length, our method has higher advantage over Wanda, empirically supporting the theoretical derivation provided in the Appendix B.1. While RIA shows improved performance with longer sequences, it also incurs more expensive time costs. Therefore, our approach offers a balanced trade-off, maintaining both accuracy and time efficiency.

