# OpenReview forum: "ACE: Exploring Activation Variance for Accurate and Calibration-Efficient LLM Pruning"
_ICLR.cc/2026/Conference — ICLR 2026 Conference Withdrawn Submission_

### Official Review · Reviewer_yHrG · 2025-10-31

**Soundness:** 2
**Presentation:** 3
**Contribution:** 2
**Rating:** 4
**Confidence:** 3

**Summary:**

The paper proposes **ACE**, a post-training pruning method combining two metrics: (i) **CosP**, a *cosine-similarity loss–guided* criterion that penalizes angular deviation between the *dense* and *pruned* layer outputs to preserve semantic orientation; and (ii) **VarP**, an *activation-variance–guided* criterion that prefers weights attached to input features with higher variance, aiming to maintain token-level distinctions under short calibration sequences. The authors claim calibration efficiency (competitive pruning quality with much shorter calibration sequences), and report results across LLaMA/LLaMA-2/OPT families for both unstructured and N:M semi-structured sparsity, with ablations on pruning time and sequence length.

**Strengths:**

- **Theoretical grounding with practical criteria.** The cosine-similarity loss and variance terms connect to semantic orientation and token-differentiation arguments, yielding implementable per-weight scores.
- **Calibration efficiency focus.** Systematic exploration of short calibration sequences (e.g., “16” tokens) shows that ACE can approach or sometimes surpass full-length Wanda/RIA in some settings, indicating reduced pruning-time and data dependence.
- **Breadth of experiments.** Evaluations span multiple model families (OPT/LLaMA/LLaMA-2) and sparsity regimes (unstructured; 2:4 & 4:8), with both LM perplexity and LM-Harness zero-shot reported.
- **Clean writing and figures.** The paper is polished, with clear tables and ablations that make replication and comparison straightforward.

**Weaknesses:**

1. **Actual speedup is unreported (Sparsity ≠ Efficiency).** The paper does not provide **end-to-end inference** metrics (latency, tokens/s, memory) under realistic sparse runtimes. Especially for unstructured sparsity, speedups depend on specialized kernels; even for N:M, practical acceleration hinges on vendor support and kernel integration. So it's not sure whether the pruned models have practical use.

2. **Significance of improvements is modest.** While pruning-time reductions under short calibration sequences are nice, pruning time is typically a one-off cost; for deployment, the decisive factors are post-pruning model quality and runtime speed. On several settings (e.g., OPT series on WikiText2), ACE’s perplexity/accuracy gains over Wanda/RIA are small; in some cases Wanda-16 is faster in pruning, and Wanda-FL can be stronger in quality. The paper would benefit from a compute-normalized Pareto (quality vs. pruning time; quality vs. sequence length) and from highlighting statistically significant wins.

3. **Minor performance gains vs. strong baselines.** For the OPT series on WikiText2, the combined method (CosP+VarP-16) often tracks or slightly improves over RIA-16, but still trails Wanda-FL in some cases and is not universally better than Wanda-16 in pruning time. The reported advantages are thin margins (fractions to ~1 perplexity point). Can you demonstrate clear wins over a stronger baseline such as SparseGPT[1] under matched calibration length and compute, especially in the practical 0–50% sparsity regime, and pair this with end-to-end speedup results on a production sparse runtime? Ultimately, post-pruning performance and speedup are the decisive metrics for deployment.

**Reference**

[1] Frantar et al. Sparsegpt: Massive language models can be accurately pruned
in one-shot.

**Questions:**

1. **Can you show real speedups/efficiency of the sparse models?**
2. **Calibration efficiency vs. quality.** Your “16-sequence” pruning reduces pruning time, but how does it trade off final model quality across tasks?

---

### Official Review · Reviewer_sj3d · 2025-10-31

**Soundness:** 4
**Presentation:** 3
**Contribution:** 2
**Rating:** 4
**Confidence:** 3

**Summary:**

The authors propose ACE (Activation Variance-guided accurate and Calibration-Efficient) pruning, an efficient post-training pruning method for LLMs. Unlike existing methods that primarily rely on the numerical magnitudes of weights and activations, ACE introduces an evaluation metric, VarP, that explores the semantic information in the input activation feature space. The core insight is that weights associated with lower input activation variance are more effective at preserving token-level semantic distinctions. The paper demonstrates that ACE can achieve high pruning accuracy while using only a small sequence length for its calibration dataset.

**Strengths:**

- Strong motivation: VarP incorporates activation variance to make better pruning decisions, with the aim of preserving token differences.

- Theoretical proof of efficiency: The authors prove why VarP yields lower error and maintains high accuracy even with short calibration sequence lengths, confirming its calibration data efficiency.

- Extensive tests and benchmarks: The authors perform tests on a wide range of models, validating the results on both unstructured and semi-structured sparsities.

**Weaknesses:**

- Weakness 1: While VarP shows some improvements, the gains are often marginal. For example, in the LLaMA-7B 50% sparsity zero-shot task, the average accuracy gain of RIA+VarP over RIA (both at Seq Len 16) is only +0.04 (55.11 vs 55.07). Similarly, for the LLaMA-13B 50% sparsity task, the gain is only +0.03 (57.31 vs 57.28).

- Weakness 2: The paper's primary speed-up claims come from comparing VarP (Seq Len 16) to Wanda (Seq Len 2048). When comparing methods at the same sequence length (16), the VarP-augmented method is often slower than the original baseline. The paper even notes that "When compared to RIA with 16 sequence lengths, VarP exhibits slightly higher latency".

- Weakness 3: The main results tables consistently omit the performance of the RIA baseline at a 2048 sequence length. This is a critical omission for evaluating the "calibration efficiency" claim, as VarP(16) is not compared against a full-sequence RIA. A single table in the appendix (Table 20) does show RIA(2048) for LLaMA-7B at 50% sparsity, and its PPL of 7.12 is actually better than Wanda+VarP(16)'s PPL of 7.18, which contradicts the paper's claims of superiority.

- Weakness 4: The use of bold values in tables in tables is inconsistent. Most tables do not use bolding to highlight the best results. Moreover, the paper would benefit from reporting mean accuracies over seeds and calculating their errors rather than representing single runs.

**Questions:**

- Question 1: The paper's central claim is "calibration efficiency" by pruning at a very short sequence length (e.g., 16). However, the models are evaluated on tasks that inherently involve much longer contexts. What is the theoretical or empirical justification for assuming that importance statistics gathered from a 16-token sequence are representative and sufficient to maintain performance when the model is later run on sequences of 2048 tokens?

- Question 2: When applying the proposed VarP metric to semi-structured sparsity (e.g., 2:4 sparsity ), is the underlying pruning mechanism identical to the baselines (just using the new VarP score), or does the method incorporate any additional techniques, such as channel-level compensation or perturbation, to manage the structural constraints?

---

### Official Review · Reviewer_jXSb · 2025-10-31

**Soundness:** 2
**Presentation:** 2
**Contribution:** 2
**Rating:** 4
**Confidence:** 4

**Summary:**

This paper proposes ACE, a post-training pruning method for LLMs. The key idea is to incorporate the variance of input activations into the weight-importance metric to preserve semantic distinctions between tokens during pruning. The authors claim this "activation variance-guided metric" (VarP) improves pruning accuracy and reduces calibration data requirements.

**Strengths:**

1. The proposed VarP metric is easy to implement and can be combined with existing pruning methods, such as Wanda and RIA, with minimal overhead.
2. Evaluations cover multiple model families and sparsity types (structured, unstructured, N:M), which demonstrates implementation effort and practical impact.

**Weaknesses:**

1. The core idea, adjusting weight importance by activation statistics, is conceptually minor. It resembles known heuristics (e.g., scaling by activation norms in Wanda) with a simple variance term added. The theoretical derivation is weak and primarily algebraic restatements of existing SparseGPT and Wanda approximations, offering limited new insight into why variance improves pruning.
2. The “calibration-efficiency analysis” is not rigorous. The derivation assumes simplistic linear relationships between variance and reconstruction error without clear assumptions or proofs. The final relation (diff $\propto$ 1 / (N + 1)) is heuristic and unverified experimentally.
3. The motivation for ``semantic distinction preservation'' is speculative and not validated. There is no analysis of token-level activation distributions or empirical evidence that variance indeed correlates with semantic collapse.
4. Reported improvements are small (≈0.1–0.3 in perplexity or 0.5–1% in accuracy) and often within measurement noise. Some results show nearly identical performance to Wanda or RIA. Without statistical tests or confidence intervals, it’s unclear whether gains are meaningful.

**Questions:**

1. Can the authors provide direct evidence that ACE preserves token-level representation diversity (e.g., cosine similarity distributions before and after pruning)?
2. How sensitive is the metric to the normalization or scaling of Xj? Since transformer activations vary across layers and tokens, how do you ensure the metric is comparable across layers?
3. The proposed metric is defined as  $S_{v a r}^{i j}=\left|W_{i j}\right| \cdot E\left[\frac{1}{1-\left\|X_j\right\|_2^2}\right]$, and is said to favor lower activation variance for “semantic distinction preservation.” Why does lower activation variance imply better preservation of semantic distinction?

---

### Note · Authors · 2025-11-26

I have read and agree with the venue's withdrawal policy on behalf of myself and my co-authors.